psychology/human-computer interaction

video game effects, media effects, loot boxes, gambling, problem gambling, adolescents

**Author for correspondence:**
David Zendle
e-mail: d.zendle@yorksj.ac.uk

# Adolescents and loot boxes: links with problem gambling and motivations for purchase

David Zendle[1], Rachel Meyer[1] and Harriet Over[2]

[1]Department of Computer Science, York St John University, 44 Lord Mayor's Walk, York, UK
[2]Department of Psychology, University of York, York, UK

DZ, 0000-0003-0279-6439

Loot boxes are items in video games that can be paid for with real-world money but contain randomized contents. Many games that feature loot boxes are played by adolescents. Similarities between loot boxes and gambling have led to concern that they are linked to the development of problem gambling in adolescents. Previous research has shown links between loot boxes and problem gambling in adult populations. However, thus far, there is no empirical evidence of either the size or existence of a link between loot box spending and problem gambling in adolescents. A large-scale survey of 16- to 18-year-olds ($n = 1155$) found evidence for such a link ($\eta^2 = 0.120$). The link between loot box spending and problem gambling among these older adolescents was of moderate to large magnitude. It was stronger than relationships previously observed in adults. Qualitative analysis of text data showed that gamers bought loot boxes for a variety of reasons. Several of these motivations were similar to common reasons for engaging in gambling. Overall, these results suggest that loot boxes either cause problem gambling among older adolescents, allow game companies to profit from adolescents with gambling problems for massive monetary rewards, or both of the above. Possible strategies for regulation and restriction are given.

## 1. Introduction

### 1.1. What are loot boxes?

Loot boxes are a relatively new way for players to spend money in video games. In recent years, loot boxes have mushroomed from a relatively obscure and unknown in-game mechanism to an industry that is predicted to generate up to $30 billion in 2018 alone [1]. There is concern among both regulators and researchers that spending money on loot boxes may be linked to gambling-related harm among both adults and children.

Making 'microtransactions' of small amounts of real-world money for virtual items or other advantages has been common in video games for many years. For example, players of the 2011 action game *Dynasty Warriors 7* can pay $0.99 to $1.99 to unlock exclusive in-game weapons; the role-playing game *Tales of Xillia* lets players pay $3 to buy 'Level Up' packs that make their characters stronger; players of the city-planning game *SimCity Buildit* can pay small amounts of money to increase the efficiency of their construction efforts. Even over a decade ago in 2005, players of the mobile game *Puzzle Pirates* could pay real-world money to buy 'doubloons', an in-game currency that could be spent on virtual items and services [2]. Similarly, in 2006, players of the open-world game *Oblivion* were able to make a microtransaction to buy cosmetic armour for their in-game horses [3].

However, in recent years, a new kind of microtransaction has become increasingly prevalent in video games: the loot box. In the examples given above, players who had paid real-world money all knew what they would get in return for this expenditure: they were buying additional levels, or doubloons, or weapons, or costumes or horse armour. By contrast, when players buy a loot box, they are not paying for something specific—they are, instead, paying for something that appears to be randomly selected from a list.

For example, in the popular first-person shooter *Counter-Strike: Global Offensive*, players can pay $2.49 to open a sealed 'weapon case'. Cases may contain extremely rare and valuable 'skins' that change the appearance of players' weapons. In fact, some of these skins are so prestigious and uncommon that they can be re-sold on secondary markets for many thousands of dollars [4]. However, when paying to open a loot box, players of *Counter-Strike* also run the risk that the case they have paid to open contains an unappealing or common item, rather than a rare or desirable one. There is no way for them to tell what they will get when they pay their money. Similarly, players of the football game *FIFA 19* pay real-world money to purchase 'player packs' that contain new footballers for their teams. These packs may contain rare and valuable players that improve their team's performance or they may not. In either case, players do not know what a pack contains when they pay real-world money for it.

## 1.2. Loot boxes and problem gambling

The behaviour outlined above involves staking real-world money on the chance outcome of a future event. Several international regulatory authorities have noted that there are striking similarities between this behaviour and gambling. This has led to various investigations across the globe of whether loot boxes in video games are in contravention of existing gambling legislation and therefore constitute an illegal and unlicensed form of gambling [5–7].

Connected to these arguments about the legal definition of loot boxes are questions about harm, and more specifically, harm to children. As the UK Gambling Commission themselves note in [8]:

> many parents are not interested in whether an activity meets a legal definition of 'gambling'. Their main concern is whether there is a product out there that could present a risk to their children.

The specific form of gambling-related harm that is most commonly associated with loot boxes is problem gambling. Problem gambling refers to disordered and excessive gambling activities that are so extreme that they lead to significant problems in an individual's personal, family and professional lives. Problem gambling is linked to factors such as depression, anxiety, bankruptcy and suicide [9–11]. One key pathway to the development of problem gambling is via conditioning: the more individuals are exposed to the arousal associated with gambling activities, the more they come to expect and require this excitement, leading to the disordered and excessive patterns of gambling-related spending mentioned above [12]. This pathway to problem gambling is considered to be a particular risk among adolescents. Indeed, exposure to gambling activities in childhood is an important predictor of problem gambling among adults [13].

There are good theoretical reasons to believe that loot boxes might be 'psychologically akin' to gambling themselves, and exposure to them might therefore cause problem gambling among children. In [14], Griffiths specifies five characteristics that differentiate gambling from other risk-related behaviours. These are:

(1) The exchange of money or something of value.
(2) A future event determines the results of this exchange, and the outcome of this event is unknown at the time that a bet is made.
(3) Chance at least partly determines the outcome of the exchange.
(4) Losses can be avoided by simply not taking part.
(5) Winners gain at the sole expense of losers.

In [15], Drummond & Sauer undertook a systematic analysis of 22 video games that feature loot boxes to determine the extent to which they fulfilled these characteristics. They found that 10 of the 22 video game loot boxes fulfilled all of the criteria listed above, and many more fulfilled most of them. They found that loot boxes not only shared 'important structural and psychological similarities with gambling', but that '100% allow for (if not actively encourage) underage players to engage with these systems'. Indeed, Drummond and Sauer concluded that the presence of loot boxes in video games might therefore be forming a 'ripe breeding ground' for the development of problem gambling among children.

Empirical research supports the existence of this effect in adult populations. In [16], Zendle & Cairns conducted a large-scale study on gamers aged 18 and over, and measured both their loot box spending and their problem gambling. Results of this study indicated that the more gamers spent on loot boxes, the more severe their problem gambling was. The effect size associated with this relationship was of magnitude $\eta^2 = 0.054$—stronger than links between problem gambling and common risk factors in the gambling literature such as drug abuse.

The correlational nature of this research meant that the researchers were unable to determine whether loot box spending caused problem gambling, or whether problem gambling among gamers made them particularly susceptible to loot box mechanisms. However, they concluded that in either case, regulation of loot boxes for harm prevention may be warranted: in one case, loot boxes are causing grave harm; in the other, they are allowing video game companies to exploit serious disorders among their customers for massive profits. Indeed, loot boxes are estimated to generate up to $30 billion dollars in 2018 alone, with this amount rising to $50 billion by 2022 [1]. This link between loot box spending and problem gambling appears robust in adult populations. Indeed, both the significance and important size of this relationship have been replicated twice already among individuals aged 18+, including in a preregistered study [17,18].

However, while links between problem gambling and loot box spending seem robust among adults, no research has yet either examined whether loot box spending is linked to problem gambling among adolescents. More importantly, no research has yet examined what the magnitude of this relationship might be. There are good reasons why loot box spending might cause problem gambling among children—and there are also good reasons why the relationship between loot box spending and problem gambling may be stronger in children than it is in the adult population.

Adolescents as a group seem particularly susceptible to problem gambling [19,20]. Indeed, problem gambling is often estimated to be more prevalent among adolescents than it is in adult populations [21–23]. There are several explanations for why adolescents might be more likely to develop problem gambling than adults. For example, neurodevelopmental research suggests that the immaturity of various aspects of brain structure and function are linked to increased impulsivity among adolescents—and this may lead to increased vulnerability to problem gambling [24,25]. Similarly, research into coping strategies among adolescents suggests that this group may lack effective ways to cope with the 'turbulent times' [26] that are associated with their time of life. They may therefore turn to gambling activities as a way to escape from painful states, leading to the development of problem gambling [27]. When taken together, these results suggest that pathways to problem gambling via loot box spending would be particularly pernicious among adolescent populations.

## 1.3. Features of loot boxes

There are a broad variety of different video games on the market. There are a similarly broad variety of different ways that loot boxes are implemented in these games. There are concerns among both academics and regulators that some kinds of loot boxes may be more strongly linked to the development of problem gambling than others [15,28]. Several key differences between kinds of loot boxes are listed below.

### 1.3.1. Some loot box contents can be cashed out

When players open loot boxes in some games, the items that they find in these boxes are bound to their account and cannot be traded with other players. Examples of games like this are *Overwatch* and *Path of Exile*. However, in other games, the items that come out of loot boxes can be traded and re-sold to other players. This had led to thriving markets for the resale of loot box items. Often, rare items are sold for significant amounts. For example, in *Counter-Strike: Global Offensive*, there is a very small chance when opening some loot boxes that a player will find a rare knife, such as the Bayonet Crimson Web. Because of their scarcity, such items can be sold to other players for large sums: the knife mentioned above, for instance, can currently be re-sold for just over $2000 [29]. Being able to 'cash out' high-value prizes from

loot boxes means that players can profit from the things that they win when making a loot box purchase. There is specific concern in the literature that loot boxes like these are more psychologically similar to other forms of gambling, and therefore more of a risk for the development of gambling-related harm [15].

### 1.3.2. Some loot box contents give gameplay advantages

Some loot boxes contain items that do not confer any gameplay advantages whatsoever. For example, loot boxes in *Team Fortress 2* and *Rocket League* only contain cosmetic items—things that change the way that players look, but do not alter their effectiveness at playing the game itself. However, this is not the case for all games. Some games have loot boxes whose contents may potentially give players a competitive edge. For example, players of *FIFA*'s Ultimate Team mode can pay real-world money to purchase 'player packs' that contain a random selection of footballers. Receiving rare and powerful footballers from these packs improves an individual's ability to play the game and is key to competing at a high level.

### 1.3.3. Some loot boxes show 'near misses'

When players open some loot boxes, they simply show the player the specific items that they have received. However, this is not the case for all loot boxes. Some loot boxes do not just show them the items that they have won, but also display 'near misses' of items that they almost seem to have won. For example, in the Multiplayer Online Battle Arena *DOTA 2*, the game displays a row of spinning rewards of varying levels of rarity and prestige. These rewards disappear one after another until only a single reward remains.

### 1.3.4. Some loot box contents can only be bought using an in-game currency

In games such as *Overwatch*, loot boxes are bought directly for cash. However, this is not the case in all games. In some games, players pay money for an in-game currency or scrip. This scrip is then used to purchase loot boxes, hiding the exact amount of real-world currency that is spent on each loot box from gamers. Examples of games with a system like this are *Fire Emblem: Heroes* and *Fortnite: Save the World*.

### 1.3.5. Some loot boxes are sometimes given away for free

In some games, loot boxes are only available for purchase, and can never be 'earned' by players for their in-game actions. An example of a game like this is *Counter-Strike: Global Offensive*, in which players must pay real-world money to purchase keys to open loot boxes. However, this is not universally the case. Many games offer players 'free' loot boxes in return for playing the game for set periods of time, levelling up or completing in-game content. For example, players of *Overwatch* receive a loot box every time they level up, and players of *League of Legends* can earn loot boxes by earning specific in-game ranks.

### 1.3.6. Some loot box contents can be 're-invested' in more loot boxes

In some games, loot boxes simply contain items or characters. However, in other games, loot boxes can contain in-game currency that can be used to purchase still more loot boxes. For example, in *Clash Royale*, players can pay an in-game currency called 'gems' to purchase or open loot boxes. These loot boxes may, in turn, contain more gems, that can be used as partial payment to purchase or open more loot boxes. Systems like this may be particularly pernicious as they may mimic the smooth playing 'to extinction' [30] that has been noted in electronic gambling machines: because players receive small amounts of currency in return for individual transactions, they may continue spending again and again without noticing that they are consistently losing capital, until no funds remain.

### 1.3.7. Some loot box contents are only available for a limited time

A final potentially important feature of loot boxes is the presence of limited time offers. Many loot boxes contain contents that are available to players for only a limited time. For example, in *Overwatch*, special edition loot boxes are available throughout the year. These contain items that are exclusively available in these loot boxes for the limited time that an in-game event runs. Similarly, in the tactical role-playing

game *Fire Emblem Heroes*, players pay to summon random heroes to join their team. However, certain 'limited heroes' can only be summoned during specific, time-limited seasonal events.

How the features described above influence the effects of loot boxes remains unclear. In [18], researchers investigated whether several specific features of different loot boxes strengthened links between loot box spending and problem gambling in an adult population. Their results provided some support for the idea that spending money on some specific kinds of loot boxes may be more strongly linked to problem gambling than others. More precisely, results suggested links between loot box spending and problem gambling may be strengthened if: (i) loot box contents could be cashed out for real-world money; (ii) loot boxes showed 'near misses' of things that gamers could have won; and (iii) the amount players spent on loot boxes was hidden behind the purchase of in-game scrip. However, the robustness of this strengthening effect remains unclear. It is also unclear whether a similar pattern of results would occur in an adolescent population. Furthermore, several potentially important differences between different kinds of loot boxes were not tested during the research outlined above. For example, neither the presence of free loot boxes, nor the presence of 're-investable' loot box contents nor the presence of 'limited time' items were investigated in previous work.

## 1.4. Why adolescents buy loot boxes

No research has yet examined why adolescents buy loot boxes. When it comes to conventional forms of gambling, a substantial body of research has highlighted that individuals engage in gambling for motivations that go far beyond simply winning money. Lloyd *et al.* [31] highlight a series of common motivations that are consistently found within the gambling literature. These consist of:

— raising money,
— excitement,
— the intrinsic enjoyment of gambling,
— escaping from stress,
— coping with adverse moods,
— a sense of inquiry and competitiveness.

The prevalence of each of these motivations varies between gambling activities. For example, research into gambling on specific casino games has suggested that they are motivated by the desire to experience a 'rush' [32], while electronic gambling machine use is associated with a need for escape [33].

Understanding why adolescents buy loot boxes may be key to understanding their effects. As noted by Neighbors *et al.* [34], identifying the motives that lie behind gambling behaviour is key to determining why problem gambling develops among some individuals. Indeed, problem gamblers often seem to have distinct reasons for gambling when compared with non-problem gamblers. In [35], researchers found that individuals who gambled in order to cope with external stresses were more likely to experience greater gambling severity, while in [33], researchers found that gambling 'to escape' was associated with gambling problems. Similarly, in both [31] and [36], researchers found that individuals who gambled with the purpose of winning money were more likely to have gambling problems.

## 1.5. Summary

Loot box spending and problem gambling are linked in adult populations. The more that adult gamers spend on loot boxes, the more severe their problem gambling is likely to be. This relationship is associated with small-to-moderate effect sizes. It appears both robust and important.

However, it is not clear how this relationship may generalize to adolescent populations. Adolescents, as a group, are particularly at risk of developing problem gambling. Indeed, the prevalence of problem gambling among adolescents is far higher than in the adult population. There are good theoretical reasons to believe that links between loot box spending and problem gambling may be stronger among adolescents than they are among adults.

In addition to this, previous research has highlighted several features of loot boxes that may strengthen relationships between loot box spending and problem gambling. However, it is not clear how reliable these strengthening effects are, and if they generalize to adolescents.

The work contained in this manuscript directly addresses the issues outlined above. It investigates both the size and importance of links between loot box spending and problem gambling in adolescents. Additionally, it clarifies whether specific features of loot boxes strengthen this link and outlines qualitative research asking adolescents why adolescents engage in loot box spending.

# 2. Methods

## 2.1. Design

We conducted a preregistered online survey with a sample of older adolescent gamers aged 16–18. Preregistration information is available at [37]. Participants were recruited via posts on reddit, a popular Internet bulletin board. Posts were made to approximately 100 'subreddits', or specialist interest bulletin boards for games that featured loot boxes.

Participants were then asked a series of questions regarding their loot box spending and problem gambling.

## 2.2. Participants

As mentioned in the preregistration information for this survey, data collection for this survey continued until a 24 h period occurred with fewer than 100 responses. Data collection began on 20 December, and ended at 00.00 on 25 December, as only 72 responses were collected on 24 December.

In total, 1158 full responses from participants aged 16–18 were collected. Two of these participants listed their monthly loot box spending at over $1 000 000. These were deemed non-serious and removed from the study. One additional participant listed their loot box spending at over $20 000 and also incorporated an abusive message to the researchers into the qualitative portion of their survey completion. They were deemed non-serious and removed from the sample. This left 1155 full responses. In addition to this, 497 responses were collected from individuals who listed their ages as numbers greater than 18. These responses were not analysed.

Participant gender was recorded via an open-response text entry box. A total of 1020 participants (88%) described themselves as 'Male' or 'M', 107 described themselves as 'Female' or 'F' (9%), and the remaining 3% of the sample identified as other categories (e.g. non-binary, genderfluid).

Three hundred and five participants (26.4%) were aged 16; 307 (26.6%) were aged 17; 543 (47.0%) were aged 18.

Overall, 687 participants (59.5%) had not paid for a loot box in the past month, and 468 participants (40.5%) had.

When it came to how quickly they had started buying loot boxes, of the 468 'buyers' in the sample, only 19 (4.1%) estimated that they had bought their first loot box within a day of playing a game; a further 20 (4.3%) estimated that they had bought a loot box within their first week of playing a game; 52 adolescents estimated they had bought their first loot box within a month of starting to play a game (11.1%); and an overwhelming majority of 377 adolescents (80.6%) estimated that they bought their first loot box more than a month after starting to play a game.

## 2.3. Measurements

Whether participants had paid to open a loot box was measured by first asking participants if they had opened a loot box in the past month via a Yes/No question, and then asking them if they had paid real-world money to open a loot box over the past month. Participants who indicated that they had done so were coded as having paid to open a loot box; all other participants were coded as not having paid to open a loot box.

Loot box spending was measured by asking participants how much money that had paid for loot boxes over the past month. This question asked participants to give their answer in the currency of the specific country that they listed as their origin in a previous question. The number given was then transformed into US dollars via the exchange rates listed in table 1 prior to analysis. All participants who had not paid to open a loot box over the past month were coded as spending $0. Spending scores were then also rank transformed prior to analysis. Previous datasets that investigate related issues (e.g. [38]) have featured extreme outliers when it comes to spending—for instance, individuals who claim to spend upwards of $2000 a month on loot boxes. To mitigate the effects of these datapoints on our inferences, rank transformation was therefore applied prior to analysis. This transformation was preregistered. The specifics of both question phrasing and transformations are listed in the preregistration document, available at [37].

Other microtransaction spending was measured by asking participants how much money that had paid for other items in games over the past month. This question asked participants to give their answer in the currency of the specific country that they listed as their origin in a previous question.

**Table 1.** Exchange rate between currencies that was used for the measurement of spending in this study.

| currency | exchange rate in US dollars |
|---|---|
| Australian dollars | 0.71 |
| Canadian dollars | 0.74 |
| Czech Koruna | 0.044 |
| Danish Krone | 0.15 |
| Euros | 1.14 |
| Hungarian Forint | 0.0036 |
| New Zealand dollars | 0.67 |
| Norwegian Krone | 0.11 |
| Polish Zloty | 0.27 |
| Pounds Sterling (£) | 1.27 |
| Romanian leu | 0.25 |
| Swedish krona | 0.11 |

The number given was then transformed into US dollars via the exchange rates listed in table 1. It was then rank transformed prior to analysis. Again, the specifics of both question phrasing and transformations are listed in the preregistration document, available at [37].

Problem gambling was measured via the Canadian Adolescent Gambling Inventory's (CAGI) [39] Problem Gambling subscale. This is a nine-item instrument that asks adolescents a series of questions about the frequency of their problem-gambling-related behaviours over the past three months. For example, one item asks 'How often have you hidden your gambling/betting from your parents, other family members or teachers?'. Participants are asked to specify how frequently they take part in behaviours by selecting one of four options: (0) never; (1) sometimes; (2) most of the time; (3) almost always. An overall measurement of problem gambling is formed from the sum of these scores, with values ranging from 0 to 27. This instrument is available at [39].

Problem gambling classification was measured by discretizing raw problem gambling scores into the following diagnostic categories: scores of 0–1 were classified as 'no problem'; scores of 2–5 were classified as 'low to moderate risk'; and scores of 6+ were classified as problem gamblers. This classification scheme forms part of the CAGI itself [39].

Impulsivity was measured via [40]. This five-item measure asks a series of Yes/No questions that relate to impulsivity (e.g. 'Do you generally do and say things without stopping to think?').

The following features of loot boxes were measured via Yes/No questions: ability to cash out, loot box contents give gameplay advantages, game gives away free loot boxes, loot box shows near misses, use of in-game currency, loot box contents are only available for a limited time and loot boxes contain items that can be 're-invested' in other loot boxes. Yes/No questions all took a similar form to the following: 'When it comes to this game, can loot box items give gameplay advantages?'. The specific wording of each of the Yes/No questions is available in the preregistration document, available at [37].

How quickly adolescents started buying loot boxes was measured by asking the following multiple-choice question: 'Approximately how long had you played this game for before paying real-world money for your first loot box in it?'. There were six possible answers: (i) less than 15 min; (ii) more than 15 min but less than an hour; (iii) more than an hour but less than a day; (iv) more than a day but less than a week; (v) more than a week but less than a month; (vi) more than a month.

Finally, motivation to buy loot boxes was measured by asking participants the following open-ended question: 'Why would you say that you buy loot boxes?'. No analysis was preregistered for this variable. A qualitative analysis of these data was instead undertaken.

## 2.4. Hypotheses

This study involves the preregistered testing of 12 specific hypotheses about the relationship between loot box spending and problem gambling. The preregistration details for these hypotheses (and all other details of this study) are available at [37]. These hypotheses (and their preregistered analysis plan) are presented below.

### 2.4.1. Hypotheses that relate to paid and unpaid openings of loot boxes

H1. There will be a significant relationship between loot box spending and problem gambling. Direction: The more a player spends on loot boxes, the more severe their problem gambling. We predict an effect size equivalent to at least $\eta^2 = 0.03$.

H2. Whether players pay money for in-game loot boxes or not will have a significant relationship with the extent of their problem gambling. Direction: Players who pay money for in-game loot boxes will have significantly higher levels of problem gambling. We predict an effect size equivalent to at least $\eta^2 = 0.03$.

H3. There will be a significant relationship between loot box spending and categorization in terms of problem gambling severity. Direction: The more a player spends on loot boxes, the more severe their categorization in terms of problem gambling severity. We predict an overall effect size equivalent to at least $\eta^2 = 0.03$.

### 2.4.2. Hypotheses that relate to specific loot box features

H4. Thinking that you are able to 'cash out' moderates the relationship between loot box spending and problem gambling. Direction: Thinking that loot boxes may be cashed out in a game will strengthen the relationship between loot box spending and problem gambling.

H5. Being able to use loot box contents for a gameplay advantage moderates the relationship between loot box spending and problem gambling. Direction: Primarily opening loot boxes in games where loot box items may give gameplay advantage will strengthen the relationship between loot box spending and problem gambling.

H6. Using an in-game currency to buy loot boxes moderates the relationship between loot box spending and problem gambling. Direction: Primarily opening loot boxes in games where loot boxes are bought with an in-game currency (that itself can be bought for real-world money) will strengthen the relationship between loot box spending and problem gambling.

H7. Showing 'near misses' in a game moderates the relationship between loot box spending and problem gambling. Direction: Primarily opening loot boxes in games where loot boxes show 'near misses' will strengthen the relationship between loot box spending and problem gambling.

H8. The presence of loot box items that are only available for a limited period of time in a game moderates the relationship between loot box spending and problem gambling. Direction: Primarily opening loot boxes in games where loot boxes are available for a limited time will strengthen the relationship between loot box spending and problem gambling.

H9. Getting currency in loot boxes that you can re-invest in further loot boxes moderates the relationship between loot box spending and problem gambling. Direction: Primarily opening loot boxes in games where loot boxes contain contents that can be re-invested will strengthen the relationship between loot box spending and problem gambling.

H10. The presence of 'free' loot boxes in a game moderates the relationship between loot box spending and problem gambling. Direction: Primarily opening loot boxes in games where loot boxes can be obtained for free as well as paid for will strengthen the relationship between loot box spending and problem gambling.

### 2.4.3. Hypotheses that relate to impulsivity

H11. There will be a significant relationship between loot box spending and impulsivity. Direction: The higher a player's impulsivity, the more they spend on loot boxes.

H12. There will be a significant relationship between problem gambling and impulsivity. Direction: The higher a player's impulsivity, the more severe their problem gambling.

## 3. Results

### 3.1. Preregistered confirmatory analyses

#### 3.1.1. Hypotheses that relate to paid and unpaid openings of loot boxes

All hypotheses were tested according to our preregistered analysis plan, available at [37]. As noted in the preregistration document, due to the high number of hypothesis tests in this study (12), Bonferroni

**Figure 1.** Problem gambling severity of gamers, split by whether they pay for loot boxes.

**Table 2.** Means and 95% confidence intervals of problem gambling, split by whether gamers pay to open loot boxes.

| loot box purchasing behaviour | problem gambling severity | N |
|---|---|---|
| gamers who do not pay to open loot boxes | 1.719 (95% CI: 1.527–1.910) | 687 |
| gamers who do pay to open loot boxes | 4.318 (95% CI: 3.858–4.778) | 468 |

corrections were applied to the results of all statistical tests, lowering the $\alpha$-level for all analyses to $p = 0.05/12$, or 0.0041.

First, H1 (There will be a significant positive correlation between loot box spending and problem gambling) was tested via calculating the Spearman rank correlation between problem gambling and loot box spending. Results indicated a significant positive correlation between loot box spending and problem gambling, supporting H1, $p < 0.001$, Spearman's $\rho = 0.347$, equivalent to $\eta^2 = 0.120$.

Next, H2 (There will be a significant relationship between whether a player pays for loot boxes and their problem gambling) was tested via a Mann–Whitney $U$-test, with Whether a player pays for loot boxes as a quasi-independent variable, and problem gambling as dependent variable. Results indicated a significant relationship between paying for loot boxes and problem gambling, supporting H2 ($U = 103\,206.500$, $p < 0.001$, $\eta^2 = 0.098$), with individuals who did not pay for loot boxes having a lower median rank and mean rank for problem gambling than those who did. A bar chart showing this relationship is depicted in figure 1. The means and 95% confidence intervals between groups are depicted in table 2.

Next, H3 (There will be a significant relationship between loot box spending and categorization in terms of problem gambling severity) was tested via a Kruskal–Wallis test with Problem gambling classification (no problem, low to moderate risk, problem gambler) as a quasi-independent variable and loot box spending as dependent variable. The effects of problem gambling (non-problem gamblers, low-risk gamblers, moderate-risk gamblers, problem gamblers) on loot box spending were tested via a Kruskal–Wallis $H$-test. Results indicated that there was a statistically significant effect of problem gambling on loot box spending, $\chi^2_{(2)} = 108.480$, $p < 0.001$, $\eta^2 = 0.119$. The means and 95% confidence intervals for each category are presented in table 3, and depicted in figure 2.

Pairwise comparisons were then conducted to measure the effects of problem gambling on loot box spending between all groups of problem gamblers via a series of six Mann–Whitney $U$-tests. These are presented in table 4.

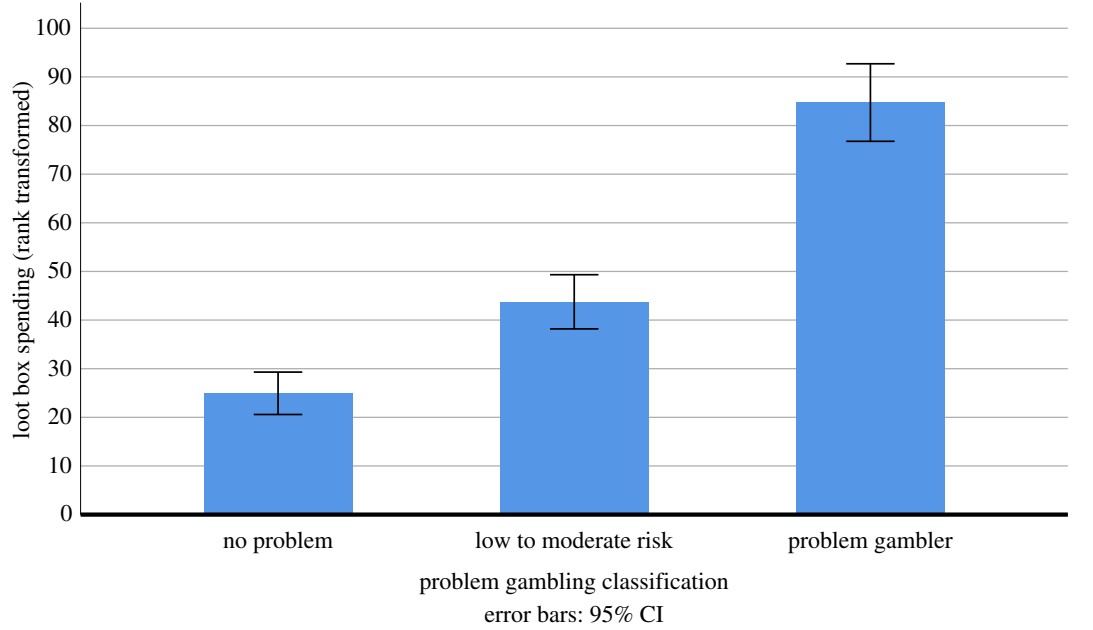

**Figure 2.** Loot box spending among older adolescents, split by problem gambling classification.

**Table 3.** Means and 95% confidence intervals for loot box spending, split by problem gambling severity. The spend statistics reported here are the mean of ranks, rather than a conversion into dollar figures. Dollar values for each relevant rank are given next to each statistic.

| | loot box spend (rank transformed) | N |
|---|---|---|
| no problem | 24.94 (rank 24: $3.64) | 604 |
| | (95% CI: 21.27 – 28.60) | |
| | rank 21: $3.36; rank 28: $4.20 | |
| low to moderate risk | 43.75 (rank 43: $6.16) | 370 |
| | (95% CI: 37.80 – 49.70) | |
| | rank 37: $5.40; rank 49: $7.10 | |
| problem gamblers | 84.72 (rank 84: $21.30) | 181 |
| | (95% CI: 74.23 – 95.22) | |
| | rank 74: $16.08; rank 95: $27.00 | |

### 3.1.2. Hypotheses that relate to specific loot box features

H4–H10 were tested via moderation analysis. Moderation analysis was run using PROCESS v. 3 for SPSS, and conducted according to [41]. Moderation was conducted under PROCESS Model 1, with $X =$ loot box spending and $Y =$ problem gambling severity in each case. The moderating variable under test, $W$, varied for each analysis. However, in each case, when relevant, 'Yes' was coded as 1 and No was coded as 0. A positive coefficient for $b_3$ is predicted in each case (i.e. the moderating variable increasing the strength of the relationship between loot box spending and problem gambling). Each moderation analysis was conducted with 10 000 bootstrap samples.

The results of these analyses are presented in table 5. Overall, the presence of only two features of loot boxes significantly strengthened links between loot box spending and problem gambling: loot box contents being available for a limited time (H8), and games giving away free loot boxes (H10).

### 3.1.3. Hypotheses that relate to impulsivity

H11 (There will be a significant relationship between loot box spending and impulsivity) was tested via calculating the Spearman rank correlation between loot box spending and impulsivity. Results showed a positive correlation between loot box spending and impulsivity. However, while they were statistically

**Table 4.** Pairwise comparisons of the effects of problem gambling on loot box spending.

| pairwise comparison groups | $U$ | $p$-value | Cohen's $d$ |
| --- | --- | --- | --- |
| no problem versus low to moderate risk | 92 171.00 | $<0.001^{a}$ | 0.298 |
| no problem versus problem gamblers | 27 322.500 | $<0.001^{a}$ | 0.783 |
| at risk versus problem gamblers | 21 827.500 | $<0.001^{a}$ | 0.590 |

[a]Effects that are significant at the $p < 0.0041$ level.

significant at the $p = 0.05$ level, they were not significant at the $\alpha$-level specified for these analyses ($p < 0.0041$). They therefore did not support H11, $p = 0.039$, Spearman's $\rho = 0.061$, equivalent to $\eta^2 = 0.003$.

H12 (There will be a significant relationship between problem gambling and impulsivity) was tested via calculating the Spearman rank correlation between problem gambling and impulsivity. Results indicated a significant positive correlation between problem gambling and impulsivity, supporting H12, $p < 0.001$, Spearman's $\rho = 0.201$, equivalent to $\eta^2 = 0.040$.

## 3.2. Non-preregistered exploratory analyses

The analyses outlined above were all preregistered in the analysis plan available at [37]. In order to explore the dataset further, we subsequently conducted a single non-preregistered exploratory analysis. We investigated the degree of association between other microtransaction spending and problem gambling, to determine if the link between loot box spending and problem gambling was uniquely strong, or if it generalized to other forms of in-game spending.

Results of a Spearman rank correlation between other microtransaction spending and problem gambling indicated a significant relationship between these variables. However, the strength of this link was almost four times smaller than the strength of the association between problem gambling and loot box spending observed in our earlier confirmatory analysis: $p < 0.001$, Spearman's $\rho = 0.180$, equivalent to $\eta^2 = 0.032$.

## 3.3. Qualitative analysis of motivation data

Only the data of individuals who had bought loot boxes in the past month ($n = 468$) were analysed. We limited the dataset in this way in order to ensure that participants' motivations were fresh in their minds and in order to make the task of conducting qualitative analysis manageable. Twenty-seven of these individuals did not state a motivation, giving a total dataset of 441 responses.

Each of these responses were first read to determine if they contained a single motivation. If an answer appeared to contain several motivations, then it was split into distinct motivations. For example, an answer to the question 'Why would you say that you buy loot boxes?' which stated 'Cosmetics and to support the developer' was split into two utterances: 'Cosmetics' and 'and to support the developer'. In total, 45 answers were deemed to contain two motivations, and three answers were deemed to contain three, giving 492 utterances in total.

Following this, each utterance was manually coded by a researcher. Codes were assigned to each utterance on the basis of the motivation it was perceived to contain. Given the exploratory approach taken here, codes were induced emergently from the data as described in [42], and were not imposed 'top down' from a pre-specified theoretical perspective. This yielded eight distinct motivations for buying loot boxes. Fifty-three utterances were deemed to contain no information about motivation and were coded as such.

In order to test the reliability of this scheme, inter-coder reliability was then calculated by having a research assistant independently code the data themselves on the basis of the coding scheme specified. The code-book used for this process is attached to this submission as appendix A.

This degree of agreement between coders was measured at Cohen's $\kappa = 0.807$, indicating excellent agreement. Each of the codes, and their prevalence within the sample, is given in table 6. A more detailed description of each motivation is also given below.

### 3.3.1. Gameplay advantages

Feel pressured to get new gear and continue to compete with ever-changing boundaries of what is classed as good gear. New gear is added constantly and thus gear quickly becomes outdated.

Enjoy the game, compete with friends. Don't want to fall behind them.

You can not be competitive in NBA2k19 or FIFA19 with out them

**Table 5.** Moderation of the relationship between loot box spending and problem gambling by various factors.

| H under test | moderating variable (W) | moderating effect of $X \times W$ on Y ($b_3$) | significance of moderating effect of $X \times W$ on Y ($b_3$) | effect of X on Y when $W = 0$ (equivalent to $b_1$) | effect of X on Y when $W = 1$ |
|---|---|---|---|---|---|
| | **confirmatory moderation analyses** | | | **associated spotlight analyses** | |
| H4 | ability to cash out | $b_3 = 0.009$ $t(1151) = 2.181$ $r^2$ change $= 0.003$ $p = 0.029^a$ | $p = 0.029^a$ | $b_1 = 0.024$ $t = 11.217$ $p < 0.001$ | 0.033 $t = 9.467$ $p < 0.001$ |
| H5 | loot box contents give gameplay advantages | $b_3 = 0.008$ $t(1151) = 1.993$ $r^2$ change $= 0.003$ $p = 0.046^a$ | $p = 0.046^a$ | $b_1 = 0.020$ $t = 6.659$ $p < 0.001$ | 0.028 $t = 11.857$ $p < 0.001$ |
| H6 | use of in-game currency to buy loot boxes | $b_3 = 0.008$ $t(1151) = 2.176$ $r^2$ change $= 0.003$ $p = 0.030^a$ | $p = 0.030^a$ | $b_1 = 0.023$ $t = 8.847$ $p < 0.001$ | 0.031 $t = 11.608$ $p < 0.001$ |
| H7 | loot boxes show 'near misses' | $b_3 = 0.010$ $t(1151) = 2.020$ $r^2$ change $= 0.003$ $p = 0.044^a$ | $p = 0.044^a$ | $b_1 = 0.024$ $t = 12.036$ $p < 0.001$ | 0.034 $t = 7.839$ $p < 0.001$ |
| H8 | loot box contents are only available for a limited time | $b_3 = 0.014$ $t(1151) = 3.079$ $r^2$ change $= 0.007$ $p = 0.002$ | $p = 0.002^b$ | $b_1 = 0.016$ $t = 4.100$ $p < 0.001$ | 0.029 $t = 14.108$ $p < 0.001$ |
| H9 | loot boxes contain items that can be 're-invested' in other loot boxes | $b_3 = 0.007$ $t(1151) = 1.602$ $r^2$ change $= 0.002$ $p = 0.110$ | $p = 0.110$ | $b_1 = 0.025$ $t = 11.316$ $p < 0.001$ | 0.031 $t = 8.845$ $p < 0.001$ |
| H10 | game gives away free loot boxes | $b_3 = 0.014$ $t(1151) = 3.002$ $r^2$ change $= 0.007$ $p = 0.003$ | $p = 0.003^b$ | $b_1 = 0.024$ $t = 12.485$ $p < 0.001$ | 0.039 $t = 8.632$ $p < 0.001$ |

[a]Moderation that is significant at the 0.05 level.
[b]Moderation that remains significant when Bonferroni corrections for the testing of 12 hypotheses are taken into account (i.e. $p < (0.05/12)$ or $p < 0.0041$).

The 21.9% of participants reported buying loot boxes to increase their in-game competitiveness. This is phrased differently for different players. Some report buying loot boxes so that they can 'keep up' with other gamers; some report buying loot boxes to enable them to 'compete' with others.

### 3.3.2. To gain specific items and characters, and to create a collection

To collect all my favorite characters, or more like a NEED to collect them all

The special limited edition items, and all the valuables that come with opening them.

Because I'm scared to miss out on the items, have a addictive need to collect them all

**Table 6.** Prevalence of motivations for buying loot boxes within the sample of older adolescents.

| motivation | frequency |
| --- | --- |
| gameplay advantages | 96 (21.9%) |
| to gain specific items and characters, and to create a collection | 84 (19.2%) |
| the fun, excitement and thrills of opening the box itself | 70 (16.0%) |
| cosmetic reasons | 67 (15.3%) |
| support the developers or pay for the game | 47 (10.7%) |
| the perception that loot boxes are good value | 43 (9.8%) |
| time advantages | 27 (6.2%) |
| profit | 4 (0.9%) |

The 19.2% of participants reported that the reason they buy loot boxes is to gain items or characters. These gamers did not specify either that these things are important because of how they look or because of any gameplay advantage attached to them. Some report that their main motivation for wanting to buy loot boxes is to create or complete a collection of things.

### 3.3.3. The fun, excitement and thrills of opening the box itself

> shit just feels good man, seeing other people opening hundreds and you get a few of that feels good and keeps me goin

> Because its addicting and thrilling reaching into the unknown

> It's not much. I never spend more than 10 dollars on a game per year really. I just do it for fun, it scratches my gambling itch.

The 16% of participants reported buying loot boxes for the experience of opening the box itself. Many described the process of opening up a loot box as fun or thrilling. A number of individuals reported buying loot boxes in order to get the 'gambling feeling' that comes with opening them.

### 3.3.4. Appearance reasons

> There's too much incentive to do so. Once you get better at the game, everyone else at your skill level has more and more expensive skins, prompting me to buy more and more to fit in

> To look better in game. To show off.

> So im a skin freak. I always liked to have skins on my tanks (WoT) i honestly dont care about the Tanks in there. So i bought them for the skins.

The 15.3% of participants reported buying loot boxes to change the way that the characters or other things they control look. These individuals may report that they buy loot boxes to 'fit in'; that they buy them to look different, look unique or look good; that they buy them to 'show off' to other players; or they may simply state that they buy loot boxes for 'cosmetics' or 'skins'.

### 3.3.5. Support the developers or pay for the game

> I like supporting developers that have a fair pay model. My general rule is that I will pay at most 1$ per hour of game time. For example, if I spend 20$ on in game currency then I wont spend any more money til I've played at least another 20 hours.

> I like to support the game that I like and this is the best option as someone who plays a lot

> I want to support game companies, but they don't want to support me.

The 10.7% of participants reported that the reason that they buy loot boxes is to support a game, or the developers who have made a game. These gamers often report that they believe that, as they are playing a 'Free to Play' game, buying loot boxes is a good way to 'give back' to the people who made this game.

### 3.3.6. The perception that loot boxes are good value

> Good value for money compared to a standard purchase

> In LoL there are skins. You can buy them or get them from loot boxes. I bought them because, if I'm lucky, I get more skins than buying them at full price.

> I calculate the odds of me getting something i'd like from a lootbox before buying them. If the odds are good enough (generally >50%), I might buy it because getting an item from the lootbox could be cheaper than buying it seperately.

The 9.8% of participants reported that the main reason that they bought loot boxes was because they perceive them to be good value. This may be because they think loot boxes cost a fair price; or it might be that loot boxes reliably contain goods or services that they believe would be more expensive to purchase elsewhere.

### 3.3.7. Time advantages

> I got really hooked on this game, not gonna lie. I wanted quick progress and they allowed me to do so.

> Progression in the game is extremely slow without loot boxes. I have been playing this game for 9 months and have normally only seen a marked progression towards end game builds after buying and opening loot boxes.

> In order to access more in-game content without having to wait for long grinds

The 6.2% of participants reported that their motivation for buying loot boxes was time. These gamers buy loot boxes in order to 'speed up' their progress through the game, or to avoid time spent 'grinding' through time-consuming portions of game content.

### 3.3.8. Profit

> Try to make profit and cashout

> For fun and to make money, I make my own money so I like to buy stuff and invest it into more skins or items that I want.

Only four participants (0.9%) reported buying loot boxes in order to make money by 'cashing out' the things that they get in the loot boxes themselves.

## 4. Discussion

### 4.1. The relationship between loot box spending and problem gambling

These results provide initial evidence for an important link between loot box spending and problem gambling in older adolescents. Whether individuals from this group paid for loot boxes or not was able to predict a significant amount of variation in their problem gambling, supporting H2. Indeed, individuals who had spent money on loot boxes within the previous month were measured as having over twice as high problem gambling severity ratings as those who had not. Similarly, correlational analyses revealed a relationship between loot box spending and problem gambling, supporting H1. Finally, non-parametric analyses revealed that problem gambling classification (no problem, low to moderate risk, problem gambler) predicted how much individuals spent on loot boxes, supporting H3.

These effects are all statistically significant even when the most stringent possible measures are taken to adjust for the testing of several hypotheses. However, their true importance lies not in their statistical significance, but in the effect sizes that are associated with them. For example, the relationship between loot box spending and problem gambling is of moderate-to-large magnitude ($\eta^2 = 0.120$) [43]. This relationship is an order of magnitude larger than relationships between problem gambling and risk factors such as alcohol dependence (equivalent $\eta^2 = 0.06$). It is of a size that commonly indicates that it bears practical, real-world significance [44]. However, most importantly, it is larger than links between problem gambling and loot box spending that have been observed in adult populations. Indeed, previous research in adults has estimated links between loot box spending and problem gambling at values ranging from $\eta^2 = 0.051$ [45] to $\eta^2 = 0.054$ [16].

This important effect size does not stand alone. All preregistered analyses of links between problem gambling and loot box spending in adolescents were associated with similar effects: the tests of H2 were associated with an effect of magnitude $\eta^2 = 0.098$; the tests of H3 were associated with effects of

magnitude $\eta^2 = 0.119$. Subgroup analyses painted an even starker picture when it came to effect sizes: for example, the difference in spending on loot boxes between those who were classified as having 'no problem' and those who were classed as 'problem gamblers' was of magnitude Cohen's $d = 0.783$, indicating an effect size that verges on being classified as large [43]. Clarifying this picture further is the fact that relationships of similar size were not seen between other microtransaction spending and problem gambling. Indeed, exploratory analyses suggested that this relationship may be substantially weaker ($\eta^2 = 0.032$) than links between loot box spending and problem gambling.

When taken together, these results clearly suggest one thing: spending money on loot boxes is linked to problem gambling in older adolescent populations. Furthermore, the severity of this relationship appears larger than in adult populations.

It is important to note that there are various reasons why this relationship might exist. To begin with, as suggested by Drummond & Sauer [15], loot boxes may provide a gateway to problem gambling. Spending money on loot boxes leads to older adolescents becoming involved in gambling activities, which may become disordered. Thus, loot box spending is linked to problem gambling because loot box spending literally causes problem gambling. This pathway to problem gambling may be particularly important in adolescent populations, given this age group's known susceptibility to developing problem gambling. This may explain the substantial effect sizes that were associated with the analyses outlined above.

However, this result may also indicate a different relationship between these variables. Loot box spending may be linked to problem gambling because problem gamblers are more likely to spend large amounts of money on loot boxes. There are good theoretical reasons to believe this, as well. As noted in [18], loot boxes share many formal features with gambling. Since problem gambling is characterized by excessive spending on gambling activities, it therefore would make sense for this excess in outlay to transfer to loot boxes as well.

This is a correlational analysis. It cannot tease apart which of these directions of causality fits our data best. Experimental and longitudinal work is desperately needed to understand these things. However, in either case, we would argue that loot boxes may be linked to social harm. In one case, loot boxes are causing problem gambling among older adolescents. In the other, they are exploiting problem gambling in this vulnerable age group in order to generate massive profits.

## 4.2. Do features of loot boxes strengthen links between loot box spending and problem gambling?

Results of a series of moderation analyses revealed that only two features of loot boxes were associated with statistically significant strengthening of links between loot box spending and problem gambling. The link between loot box spending and problem gambling was strengthened in games where loot box items were only available for a limited time. It was also strengthened in cases where games would occasionally give away 'free' loot boxes.

The precise reason for these effects is unclear. For example, it may be the case that 'limited time' offers on loot box items create a sense of urgency that appeals to problem gamblers more than it does those with no problem. This would lead to problem gamblers spending more money in these games than those with no problem, strengthening links between loot box spending and problem gambling. Similarly, games that give away free loot boxes might offer older adolescents who are susceptible to developing problem gambling a 'taste' of a gambling-like mechanism, leading to increased associations between problem gambling and loot box spending for players of those games. However, it is important to interpret the effects seen above with caution. To begin with, while it was statistically significant, the effect size associated with each relationship was very small: an $r^2$ change of 0.007. In other words, each of these features was only able to account for an additional 0.7% of problem gambling among players. The real-world importance of these effects is therefore unclear. Thus, while the results outlined above highlight the potential importance of in-game features, further experimental work is necessary to determine their precise relationship with problem gambling.

## 4.3. Links between impulsiveness and loot box spending

Adolescents are thought to be particularly susceptible to developing problem gambling due to a variety of factors. One of these factors is impulsiveness, which is linked to immature neurodevelopment. In this study, consonant with this perspective (and H12), problem gambling among adolescents was linked to impulsiveness ($\eta^2 = 0.040$).

However, contrary to our hypotheses, there was no significant link between impulsiveness and loot box spending ($\eta^2 = 0.003$). The reasons for this lack of a link are unclear. However, it suggests that pathways to loot box spending among older adolescents may be quite different to pathways to problem gambling in this group. It seems probable that a variety of factors other than impulsiveness provide more useful explanations for adolescent spending on loot boxes. For example, loot boxes commonly contain 'cosmetic' items, or 'skins'. It may therefore be the case that social pressure, or a desire to appear a certain way to other players, is a key determinant of loot box spending. Further qualitative research into the motivations of adolescents when it comes to buying loot boxes is necessary in order to better understand this.

## 4.4. Motivations for loot box spending

Several motivations for spending money on loot boxes seem analogous to reasons for gambling. For example, 70 of the individuals in our sample explicitly mentioned that they opened loot boxes in order to get the exciting feeling that came from opening them. Some explicitly linked this to the randomized nature of the rewards in loot boxes; others even referred to these as 'gambling feeling[s]'. There are parallels that can be drawn here between why older adolescents buy loot boxes and why gamblers spend money on games like Internet poker, in which excitement is often thought to be a key factor in determining engagement [46].

However, just as striking as similarities between motivations are differences. Only four motivations from a pool of 492 mentioned the idea of 'profit'. This contrasts heavily with gambling, in which the desire to make money is often a key driver of why many people gamble. This may point to the idea that the value associated with loot boxes lies somewhere other than in financial terms. Indeed, the most common motivation for spending money on loot boxes within our sample was for gameplay advantages: loot boxes were bought not because they might increase players' capital outside of the game, but because they might enable them to compete within the game itself.

A similar point may be made about social capital. Sixty-seven responses referred to the idea that players were buying loot boxes in the hope that they might allow them to look a certain way. Some even mentioned the idea that the cosmetic items and skins contained in loot boxes were necessary to let them 'fit in' within a social group. It may be the case that the desire for social acceptance within a group is a key driver of loot box spending. A key priority for future research is to determine whether this motivation is particularly important for adolescent gamers.

## 4.5. Limitations and further work

As noted above, the primary limitation of this study is its correlational nature. Further work is necessary to understand the nature of relationship between loot box spending and problem gambling. In order to better discern the causal direction of this relationship, both longitudinal and experimental research is needed.

This research is also limited by the composition of its sample. To begin with, the age group under test in this study consisted entirely of older adolescents aged between 16 and 18. Replication studies are needed to determine whether the observed effects replicate with younger participants. In addition, participants were recruited from the online bulletin-board reddit and elected to take part in a study that investigated loot boxes. It is possible that results from this sample would not generalize to the wider population. However, it is important to note that previous loot box research has used samples of this nature (i.e. [16]), and results have been shown to replicate well to other contexts [18,45]. Nevertheless, in order to establish the generalizability and robustness of the effects seen here, further research using more representative samples of adolescent gamers would be valuable. Finally, we did not have a mechanism in place that enabled us to systematically screen for inattentive or careless responding. However, the focused responses to the open-ended questions and the level of detail provided by participants in their responses suggest that participants took the study seriously. A final note must be made regarding gaming disorder: a proposed pathological condition which is typically conceptualized as a situation in which an individual's engagement with gaming becomes so excessive that it causes serious problems for the gamer and those around them. The progression of disordered gaming is often modelled in a similar fashion to the development of disordered gambling. Due to similarities between loot boxes and gambling, it may seem possible to interpret our results as providing important evidence for the existence of gaming disordered as conceptualized above. This is not the case. Indeed, this research provides no evidence that spending on loot boxes is linked to excessive gaming in any way. Further work is necessary to establish if this is the case.

# 5. Conclusion

The more money that older adolescents spent on loot boxes, the greater their problem gambling severity. Older adolescents who spent money on loot boxes displayed more than twice as high measurements of problem gambling than those who did not. Adolescent problem gamblers spent more than five times as much money on loot boxes than those who did not have a problem. Problem gambling and loot box spending were linked by an association of magnitude $\eta^2 = 0.120$: more than twice as strong as the relationship seen recently in a similarly recruited adult population.

There is one clear conclusion that can be drawn from these results: when video game companies allow adolescents to buy loot boxes, they are potentially exposing them to negative consequences. It may be the case that loot box spending in adolescents causes problem gambling. It may be the case that loot boxes allow games companies to monetize problem gambling in these vulnerable populations for 11-digit annual profits. We believe that both relationships may potentially lead to serious adverse consequences for younger gamers.

Loot boxes may have generated up to $30 billion in 2018 [1]. It is unclear how much of this revenue has come from adolescents. We would argue that regardless of the profitability of the loot box trade, the risks associated with them are worryingly high.

There are a broad range of decisions that interested parties can make in order to minimize any risks associated with loot boxes. Ratings agencies may consider restricting access to games with paid loot boxes to players who are of legal gambling age. Alternatively, they may consider attaching content descriptors to games which feature loot boxes in order to ensure that parents and gamers are able to make an informed choice when purchasing a game that features loot boxes. Even with adults, games companies may consider implementing responsible spending provisions in games that feature loot boxes. For example, they may consider implementing the ability for players to voluntarily set limits on the extent of their loot box spending [47].

Finally, national and federal authorities may consider regulating loot boxes in the same way that they would if they fulfilled the technical requirements necessary to be considered a form of gambling.

Ethics. Ethical approval for this research was granted by the York St John University Cross-Departmental Ethics Board, under submission code 2154/2155. Informed consent was gathered from each participant. The full script for this study is available at the OSF repository detailed below.

Data accessibility. All datasets that are relevant to this study are freely available from the OSF repository located at https://osf.io/ts7ue/. Identifier: doi:10.17605/OSF.IO/TS7UE.

Authors' contributions. D.Z. created the initial study design, with assistance from H.O. in the inclusion and measurement of specific variables. R.M. and D.Z. ran the survey together, and D.Z. provided first-coding of qualitative data from it. R.M. provided second coding of these data. D.Z. drafted the initial paper, and R.M. and H.O. provided critique and feedback on it. H.O. provided final edits to the paper. All authors have read and approved this manuscript.

Competing interests. We have no competing interests.

Funding. No funding supported this research.

Acknowledgements. No further parties contributed to this study but did not meet criteria for authorship.

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
