## [Reviewer comments · Royal Society Open Science]

Review History

RSOS-190049.R0 (Original submission)

Review form: Reviewer 1 (Aaron Drummond)

Is the manuscript scientifically sound in its present form?

Yes

Are the interpretations and conclusions justified by the results?

Yes

Is the language acceptable?

Yes

Is it clear how to access all supporting data?

Yes

Do you have any ethical concerns with this paper?

No

Have you any concerns about statistical analyses in this paper?

Yes

Recommendation?

Accept with minor revision (please list in comments)

Comments to the Author(s)

The present study investigates the relationship between spending on video game loot boxes among both adults and a limited adolescent population. The paper investigates an important and timely issue and follows many best practices in the research. The findings show that the relationship between loot box spending and problem gambling observed by Zendle and Cairns (2018) is replicable, and appears to be larger in a sample of adolescents. There is much to like about this piece of research. The authors have preregistered the study, obtained a sizeable sample, adjusted their p values for multiple comparisons appropriately, and attempted to answer some important questions. For the most part, they have achieved this well. However, there are some changes to the paper I would like to see implemented before the paper is published.

The first is the way that the authors discuss the adolescent population that responded to the sample. Throughout the manuscript, the authors refer to the effects for the adolescent population, however, they have not obtained a suitable sample to make such inferences. The actual demographics for the current study are ages 16-18, which are late stage adolescents. While this is likely due to ethical considerations it does limit the inferences we can make about the younger adolescent population. Although theory would predict that younger adolescents would exhibit a stronger relationship than older adolescents, it is difficult to make statements such as "problem gambling amongst adolescents was more than twice as strong as the relationship observed in adults", when only older adolescents are sampled. I suggest that the authors carefully edit their manuscript to make it clear that the effects they report are specifically for older adolescents.

The sample the authors have used for this study is also worthy of comment. Samples recruited from Reddit are not exactly random samples of the population of interest, and there is now considerable public interest and debate about the issue of loot boxes. Although the authors have indicated some data cleaning has occurred, I would like to know more about this. For instance, did the researchers employ any attention checks as are common when using online samples? Did the authors screen for any trolling responses (e.g., Zendle and Cairns' 2018 data contained a non-trivial number of participants who reported Apache attack helicopter, a common internet meme as their gender). Do the results hold up if such responses are removed? It would be nice to see that the effects are present irrespective of analysis strategies. I would also like to see the authors comment on the sample in the discussion section as one of the limitations of the present paper.

Another issue in the manuscript is the use of effect sizes and the way that the authors discuss them. The authors use partial eta squared, which is not a particularly useful effect size for interpreting the size of relationships and is notorious for being influenced by issues such as sample size. It would be preferable if the authors could adopt a standardized effect size measure throughout the manuscript to aid the reader in interpreting the size of the reported effects. Additionally, there is a tendency for the authors to discuss effect sizes as a "number of times" larger than other effect sizes. This is inadvisable as the effect size one uses will alter the relationship between effect sizes. For instance it would be quite common for authors to report the percentage of variance explained (r^2) in study like this, and the relationship between two r^2 scores will diverge more than their r scores. Similarly, orders of magnitude aren't particularly informative in any case, because an r of .02 is twice the size of an r of .01, yet both are utterly meaningless. For these reasons, it would probably be better for the authors to interpret the difference in effect sizes as not an order of magnitude, but simply as differences in the size of the effect.

A very minor point, but page 19 has the word “7high” on it.

Overall, I believe the manuscript is worthwhile and with some relatively minor editing and clarification should make a good addition to the literature.

Review form: Reviewer 2 (Christopher Ferguson)

Is the manuscript scientifically sound in its present form?

Yes

Are the interpretations and conclusions justified by the results?

Yes

Is the language acceptable?

Yes

Is it clear how to access all supporting data?

Yes

Do you have any ethical concerns with this paper?

No

Have you any concerns about statistical analyses in this paper?

No

Recommendation?

Accept with minor revision (please list in comments)

Comments to the Author(s)

The current study is a preregistered survey design of loot-box spending and problematic gambling in a large sample of adolescents. I thought the article had many positive qualities. Contingent upon some revisions, I believe it would be acceptable for publication.

First the authors did a very good and comprehensive job with the literature. However, I thought maybe it could be shortened a bit. I know they touch upon a lot of issues, but I might suggest dialing back on policy issues in particular (and again in the discussion) and sticking to the facts rather than prescriptions. Overall I think shortening the lit review by maybe 25% would help the readability of the paper.

Naturally it would be inappropriate for me to suggest shortening the lit review without also suggesting an addition. This is just a suggestion, but by focusing on problematic gambling, the authors dodge the very contentious problematic gaming debate. Maybe that's for the best, which is why I say this is just a suggestion, although I suspect there are some who will use this article anyway to make points about problematic gaming. I wonder if it's worth a few sentences at least to nip that particular issue in the bud. I kind of like the equating of loot boxes specifically to pathological gambling but would hate for others to ignore the authors' caution here. The authors could point out this is a unique issue for some games, and should not be used to make assertions about all games.

I think the article's main weaknesses is the lack of reliability check items for careless or mischievous responding. Something to consider in future research, but maybe just a note for limitation for now.

The statistical analyses are all competently done as far as I can see. It's great they preregistered their design!

Last, I'd suggest dialing back the policy recommendations in the discussion. I generally don't advise making policy recommendations (which suggest causality) from any kind of correlational data. Similarly I'd argue against using words like "harm" ...try to keep the language conservative and cautious and let the data speak for itself. I think too many media psychologists have prematurely used the word "harm" in the past for that particular bell to have a remotely pleasant ring anymore, however good-faith the authors here are using it.

Signed,
Chris Ferguson

Decision letter (RSOS-190049.R0)

24-Apr-2019

Dear Dr Zendle

On behalf of the Editors, I am pleased to inform you that your Manuscript RSOS-190049 entitled "Adolescents and loot boxes: Links with problem gambling and motivations for purchase" has been accepted for publication in Royal Society Open Science subject to minor revision in accordance with the referee suggestions. Please find the referees' comments at the end of this email.

The reviewers and handling editors have recommended publication, but also suggest some minor revisions to your manuscript. Therefore, I invite you to respond to the comments and revise your manuscript.

- Ethics statement

- Data accessibility

If you wish to submit your supporting data or code to Dryad (<http://datadryad.org/>), or modify your current submission to dryad, please use the following link:
<http://datadryad.org/submit?journalID=RSOS&manu=RSOS-190049>

- **Competing interests**

- **Authors' contributions**

- **Acknowledgements**

- **Funding statement**

Because the schedule for publication is very tight, it is a condition of publication that you submit the revised version of your manuscript before 03-May-2019. Please note that the revision deadline will expire at 00.00am on this date. If you do not think you will be able to meet this date please let me know immediately.

When submitting your revised manuscript, you will be able to respond to the comments made by the referees and upload a file "Response to Referees" in "Section 6 - File Upload". You can use this to document any changes you make to the original manuscript. In order to expedite the

processing of the revised manuscript, please be as specific as possible in your response to the referees. We strongly recommend uploading two versions of your revised manuscript:

Kind regards,
Andrew Dunn
Royal Society Open Science Editorial Office
Royal Society Open Science

on behalf of Dr Anastasia Christakou (Associate Editor) and Essi Viding (Subject Editor)
openscience@royalsociety.org

Reviewer comments to Author:

Reviewer: 1

Comments to the Author(s)

The present study investigates the relationship between spending on video game loot boxes among both adults and a limited adolescent population. The paper investigates an important and timely issue and follows many best practices in the research. The findings show that the relationship between loot box spending and problem gambling observed by Zendle and Cairns (2018) is replicable, and appears to be larger in a sample of adolescents. There is much to like about this piece of research. The authors have preregistered the study, obtained a sizeable sample, adjusted their p values for multiple comparisons appropriately, and attempted to answer some important questions. For the most part, they have achieved this well. However, there are some changes to the paper I would like to see implemented before the paper is published.

The first is the way that the authors discuss the adolescent population that responded to the sample. Throughout the manuscript, the authors refer to the effects for the adolescent population, however, they have not obtained a suitable sample to make such inferences. The actual demographics for the current study are ages 16-18, which are late stage adolescents. While this is likely due to ethical considerations it does limit the inferences we can make about the younger adolescent population. Although theory would predict that younger adolescents would exhibit a stronger relationship than older adolescents, it is difficult to make statements such as “problem gambling amongst adolescents was more than twice as strong as the relationship observed in adults”, when only older adolescents are sampled. I suggest that the authors carefully edit their manuscript to make it clear that the effects they report are specifically for older adolescents.

The sample the authors have used for this study is also worthy of comment. Samples recruited from Reddit are not exactly random samples of the population of interest, and there is now considerable public interest and debate about the issue of loot boxes. Although the authors have indicated some data cleaning has occurred, I would like to know more about this. For instance, did the researchers employ any attention checks as are common when using online samples? Did the authors screen for any trolling responses (e.g., Zendle and Cairns’ 2018 data contained a non-trivial number of participants who reported Apache attack helicopter, a common internet meme as their gender). Do the results hold up if such responses are removed? It would be nice to see that the effects are present irrespective of analysis strategies. I would also like to see the authors comment on the sample in the discussion section as one of the limitations of the present paper.

Another issue in the manuscript is the use of effect sizes and the way that the authors discuss them. The authors use partial eta squared, which is not a particularly useful effect size for interpreting the size of relationships and is notorious for being influenced by issues such as sample size. It would be preferable if the authors could adopt a standardized effect size measure throughout the manuscript to aid the reader in interpreting the size of the reported effects. Additionally, there is a tendency for the authors to discuss effect sizes as a “number of times” larger than other effect sizes. This is inadvisable as the effect size one uses will alter the relationship between effect sizes. For instance it would be quite common for authors to report the percentage of variance explained (r^2) in study like this, and the relationship between two r^2

scores will diverge more than their r scores. Similarly, orders of magnitude aren't particularly informative in any case, because an r of .02 is twice the size of an r of .01, yet both are utterly meaningless. For these reasons, it would probably be better for the authors to interpret the difference in effect sizes as not an order of magnitude, but simply as differences in the size of the effect.

A very minor point, but page 19 has the word "7high" on it.

Overall, I believe the manuscript is worthwhile and with some relatively minor editing and clarification should make a good addition to the literature.

Reviewer: 2

Comments to the Author(s)

The current study is a preregistered survey design of loot-box spending and problematic gambling in a large sample of adolescents. I thought the article had many positive qualities. Contingent upon some revisions, I believe it would be acceptable for publication.

First the authors did a very good and comprehensive job with the literature. However, I thought maybe it could be shortened a bit. I know they touch upon a lot of issues, but I might suggest dialing back on policy issues in particular (and again in the discussion) and sticking to the facts rather than prescriptions. Overall I think shortening the lit review by maybe 25% would help the readability of the paper.

Naturally it would be inappropriate for me to suggest shortening the lit review without also suggesting an addition. This is just a suggestion, but by focusing on problematic gambling, the authors dodge the very contentious problematic gaming debate. Maybe that's for the best, which is why I say this is just a suggestion, although I suspect there are some who will use this article anyway to make points about problematic gaming. I wonder if it's worth a few sentences at least to nip that particular issue in the bud. I kind of like the equating of loot boxes specifically to pathological gambling but would hate for others to ignore the authors' caution here. The authors could point out this is a unique issue for some games, and should not be used to make assertions about all games.

I think the article's main weaknesses is the lack of reliability check items for careless or mischievous responding. Something to consider in future research, but maybe just a note for limitation for now.

The statistical analyses are all competently done as far as I can see. It's great they preregistered their design!

Last, I'd suggest dialing back the policy recommendations in the discussion. I generally don't advise making policy recommendations (which suggest causality) from any kind of correlational data. Similarly I'd argue against using words like "harm" ...try to keep the language conservative and cautious and let the data speak for itself. I think too many media psychologists have prematurely used the word "harm" in the past for that particular bell to have a remotely pleasant ring anymore, however good-faith the authors here are using it.

Signed,
Chris Ferguson

Author's Response to Decision Letter for (RSOS-190049.R0)

See Appendix A.

Decision letter (RSOS-190049.R1)

10-May-2019

Dear Dr Zendle,

I am pleased to inform you that your manuscript entitled "Adolescents and loot boxes: Links with problem gambling and motivations for purchase" is now accepted for publication in Royal Society Open Science.

You can expect to receive a proof of your article in the near future. Please contact the editorial office (openscience_proofs@royalsociety.org and openscience@royalsociety.org) to let us know if you are likely to be away from e-mail contact. Due to rapid publication and an extremely tight schedule (publication usually follows in 4-6 weeks), if comments are not received, your paper may experience a delay in publication.

on behalf of Dr Anastasia Christakou (Associate Editor) and Essi Viding (Subject Editor)
openscience@royalsociety.org

Appendix A

Dear Dr. Christakou and Professor Viding,

We would like to thank the reviewers and editors for their comments.

We found them to be both rigorous and fair. We have tried to address all of the issues raised and provide a point-by-point response below.

General comments

Our manuscript incorporates statements describing the ethical approval that was received for this study as well as details of how informed consent was obtained. See subsections 'Research Ethics'; 'Animal Ethics'.

We have ensured that our manuscript contains a data availability section that is consonant with the requirements outlined above. See manuscript subsection 'Data Availability'.

We have no competing interests and have declared this. We have declared our input to the manuscript in the format suggested. We have declared the source of funding for the research. See subsections 'Competing interests' and 'Funding'.

R1 Comments

However, there are some changes to the paper I would like to see implemented before the paper is published.

The first is the way that the authors discuss the adolescent population that responded to the sample. Throughout the manuscript, the authors refer to the effects for the adolescent

population, however, they have not obtained a suitable sample to make such inferences. The actual demographics for the current study are ages 16-18, which are late stage adolescents. While this is likely due to ethical considerations it does limit the inferences we can make about the younger adolescent population. Although theory would predict that younger adolescents would exhibit a stronger relationship than older adolescents, it is difficult to make statements such as “problem gambling amongst adolescents was more than twice as strong as the relationship observed in adults”, when only older adolescents are sampled. I suggest that the authors carefully edit their manuscript to make it clear that the effects they report are specifically for older adolescents.

In line with R1's suggestion, we have adjusted the wording of the manuscript to make it clear that the sample under test here is specifically formed of older adolescents. We have also added to the discussion a call for further research into other age groups (e.g. to investigate whether these effects replicate with younger adolescents).

The sample the authors have used for this study is also worthy of comment. Samples recruited from Reddit are not exactly random samples of the population of interest, and there is now considerable public interest and debate about the issue of loot boxes. Although the authors have indicated some data cleaning has occurred, I would like to know more about this. For instance, did the researchers employ any attention checks as are common when using online samples? Did the authors screen for any trolling responses (e.g., Zendle and Cairns' 2018 data contained a non-trivial number of participants who reported Apache attack helicopter, a common internet meme as their gender). Do the results hold up if such responses are removed? It would be nice to see that the effects are present irrespective of analysis strategies. I would also like to see the authors comment on the sample in the discussion section as one of the limitations of the present paper.

In line with R1's suggestion, we have extended the discussion section of our manuscript to draw attention to the limitations of a self-selected sample, and to call for further research that uses representative samples of adolescents.

An overview of answers to gender questions within the sample revealed only 4 responses out of 1155 that might be considered non-serious: Two participants listed their gender as 'Apache attack helicopter'; one participant listed their gender as 'dude'; one participant listed their gender as 'dragon'. The analyses outlined above were re-run with these participants excluded from the sample. There was no change in the significance of any statistical test with this reduced sample.

Another issue in the manuscript is the use of effect sizes and the way that the authors discuss them. The authors use partial eta squared, which is not a particularly useful effect size for interpreting the size of relationships and is notorious for being influenced by issues such as sample size. It would be preferable if the authors could adopt a standardized effect size measure throughout the manuscript to aid the reader in interpreting the size of the reported effects. Additionally, there is a tendency for the authors to discuss effect sizes as a “number of times” larger than other effect sizes. This is inadvisable as the effect size one uses will alter the relationship between effect sizes. For instance it would be quite common for authors to report the percentage of variance explained (r^2) in study like this, and the relationship between two r^2 scores will diverge more than their r scores. Similarly, orders of magnitude aren't particularly informative in any case, because an r of .02 is twice the size of an r of .01, yet both are utterly meaningless. For these reasons, it would

probably be better for the authors to interpret the difference in effect sizes as not an order of magnitude, but simply as differences in the size of the effect.

In line with R1's suggestions, we have made adjustments throughout the manuscript to our discussion of variance-explained measures of effect size. More specifically, we have revised any occasions where we refer to effect sizes as being 'multipliers' of other effect sizes and have instead, as suggested, referred to differences between these effect sizes.

A very minor point, but page 19 has the word "7high" on it.

We have adjusted the manuscript to remove this error.

R2 Comments

First the authors did a very good and comprehensive job with the literature. However, I thought maybe it could be shortened a bit. I know they touch upon a lot of issues, but I might suggest dialing back on policy issues in particular (and again in the discussion) and sticking to the facts rather than prescriptions. Overall I think shortening the lit review by maybe 25% would help the readability of the paper.

In line with R2's suggestion, we have contracted the literature review by shortening our summary of policy implications. We have rephrased this content in the discussion in line with R2's other comments.

Naturally it would be inappropriate for me to suggest shortening the lit review without also suggesting an addition. This is just a suggestion, but by focusing on problematic gambling, the authors dodge the very contentious problematic gaming debate. Maybe that's for the best, which is why I say this is just a suggestion, although I suspect there are some who will use this article anyway to make points about problematic gaming. I wonder if it's worth a few sentences at least to nip that particular issue in the bud. I kind of like the equating of loot boxes specifically to pathological gambling but would hate for others to ignore the authors' caution here. The authors could point out this is a unique issue for some games, and should not be used to make assertions about all games.

In line with R2's suggestion, we have extended our discussion to incorporate a brief passage about links between our research and disordered gaming.

I think the article's main weaknesses is the lack of reliability check items for careless or mischievous responding. Something to consider in future research, but maybe just a note for limitation for now.

As suggested by R2, we have extended our manuscript to incorporate a discussion of this limitation.

Last, I'd suggest dialing back the policy recommendations in the discussion. I generally don't advise making policy recommendations (which suggest causality) from any kind of correlational data. Similarly I'd argue against using words like "harm"...try to keep the language conservative and cautious and let the data speak for itself. I think too many media psychologists have prematurely used the word "harm" in the past for that particular bell to have a remotely pleasant ring anymore, however good-faith the authors here are using it.

In line with R2's comments, we have revised the wording of our discussion. We have also nuanced our discussion of policy.